# Population history and genome wide association studies of birth weight in a native high altitude Ladakhi population

**Sushil Bhandari**[1], **Padma Dolma**[2], **Mitali Mukerji**[3], **Bhavana Prasher**[3], **Hugh Montgomery**[4], **Dalvir Kular**[5], **Vandana Jain**[6], **Vatsla Dadhwal**[6], **David J. Williams**[7], **Aniket Bhattacharyaa**[3], **Edmund Gilbert**[1], **Gianpiero L. Cavalleri**[1], **Sara L. Hillman**[7] *

1 Royal College of Surgeons in Ireland, Dublin, Ireland, 2 Sonam Norboo Memorial Hospital, Leh, Ladakh, India, 3 Council Scientific Industrial Research-Institute for Genomics and Integrative Biology, New Delhi, India, 4 Centre for Human Health and Performance, University College London, London, United Kingdom, 5 King's College Hospital, London, United Kingdom, 6 All India Institute for Medical Sciences, New Delhi, India, 7 University College London Institute for Women's Health, London, United Kingdom

* sara.hillman@ucl.ac.uk

**Data Availability Statement:** All relevant data are within the manuscript and its Supporting Information files.

## Abstract

Pathological low birth weight due to fetal growth restriction (FGR) is an important predictor of adverse obstetric and neonatal outcomes. It is more common amongst native lowlanders when gestating in the hypoxic environment of high altitude, whilst populations who have resided at high altitude for many generations are relatively protected. Genetic study of pregnant populations at high altitude permits exploration of the role of hypoxia in FGR pathogenesis, and perhaps of FGR pathogenesis more broadly. We studied the umbilical cord blood DNA of 316 neonates born to pregnant women managed at the Sonam Norboo Memorial Hospital, Ladakh (altitude 3540m) between February 2017 and January 2019. Principal component, admixture and genome wide association studies (GWAS) were applied to dense single nucleotide polymorphism (SNP) genetic data, to explore ancestry and genetic predictors of low birth weight. Our findings support Tibetan ancestry in the Ladakhi population, with subsequent admixture with neighboring Indo-Aryan populations. Fetal growth protection was evident in Ladakhi neonates. Although no variants achieved genome wide significance, we observed nominal association of seven variants across genes (*ZBTB38*, *ZFP36L2*, *HMGA2*, *CDKAL1*, *PLCG1*) previously associated with birthweight.

## Introduction

Birth weight is a complex trait driven by metabolic, vascular and immune interactions between mother and fetus. Successful placental and fetal growth leads to appropriate birth weight and reduced neonatal morbidity, ultimately driving success of a species [1–3]. The inability for a fetus to reach its growth potential (fetal growth restriction, FGR) compromises neonatal survival [1, 4–6]. A recent Delphi consensus further defines FGR on the basis of severity of growth restriction (i.e. < 3rd centile, or association of a birthweight <10th centile and concomitant changes in fetal hemodynamics as defined by doppler ultrasound of blood flow) [7].

**Funding:** SH was funded for this work by a Wellcome Trust SEED trust award (WT 109862/Z/15/Z). A UCL Global Health Challenges award was received by HM and SH and used for travel expenses. The Royal College of Obstetricians and Gynaecologists also awarded the Eden Travelling Fellowship to SH to develop the relationship with the maternity team at Sonam Norboo Memorial Hospital. HM and DJW supported by the UK National Institute for Health Research's Comprehensive Biomedical Research Centre at University College London Hospitals.SB was supported by a Government of Ireland Postdoctoral Fellowship from the Irish Research Council (GOIPD/2018/408). The funders had no role in study design, data collection and analysis, decision to publish, or preparation of the manuscript

**Competing interests:** The authors have declared that no competing interests exist.

Studies support a genetic influence on birth weight [8]. Genome wide association studies (GWAS) have identified approximately 70 single nucleotide polymorphisms (SNPs) of robust influence [9], although these studies have primarily related to European ancestry cohorts of normal birth weight. Exploring the genetic contribution to *in-utero* fetal growth, particularly in populations where evolutionary selection may have resulted in protected birth weight from negative environmental influences, has the potential to enhance understanding of FGR pathogenesis, and thus to suggest new protective strategies.

Although FGR is more prevalent in those residing above 1500m (high altitude, where oxygen availability is reduced) [4, 10, 11], populations who have resided at high altitude for many generations appear to be protected [12, 13]. For example, babies born to (native ancestral high altitude) Tibetan women at altitudes above 3000m are more than 500g heavier than those born to native lowland Han Chinese [14]. Similarly, in La Paz, Bolivia (3,600m), native Andean babies are born heavier than their European counterparts [15].

To date, genetic studies of birth weight at high altitude have primarily focused on maternal genotypes. The maternal *PRKAA1* gene locus (coding for *AMPK*, a central regulator of cellular energy metabolism) has been associated with birth weight and maternal uterine artery diameter in high altitude Andean residents [16]. Less is known about fetal genotype and its association with birth weight at high altitude in Tibetan ancestry populations.

Ladakh, in the Jammu and Kashmir region of India, lies between the Karakoram and Himalayan mountain ranges. The term "La-dvags" in Tibetan means "land of high passes" and "Ladakh" is the Persian spelling [17]. The highland Ladakh region connects South Asia with the Tibetan plateau via the ancient trade corridor- "the Silk Road". Genetic adaptation to the hypoxia of high altitude is well described [18]. Although sharing similar linguistic, cultural and religious practices with Tibetans [17, 19], the population structure and genetic selectionof the Ladakhi population is relatively poorly studied. The Sonam Norboo Memorial Hospital in Leh [3,540m], which provides maternity care for Ladakh, has an institutional birth rate of >90% [20] making it a unique site to study a pregnant population at high altitude.

We hypothesized that the fetal genotype of native Ladakhis would be enriched for gene variants which protect *in-utero* growth against the challenging environment of high altitude gestation. We first performed a population genetic survey before seeking to identify fetal genetic elements associated with birth weight in the Ladakhi population.

## Materials and methods

Ethical permissions were granted for the study from the Indian Health Ministry's Screening committee (HMSC, 7th September 2016), the Office of the Chief Medical Officer Leh (3rd August 2016), the All India Institute for Medical Sciences, and the University College London Research Ethics Committee (3634/002). All methods were performed in accordance with the relevant guidelines and regulations and informed consent was obtained from all participants and/or their legal guardians. The research was performed in accordance with the Declaration of Helsinki. Pregnant women and their partners were recruited from Sonam Norboo Memorial (SNM) Hospital, Leh [altitude 3,540m] from antenatal clinics and the ultrasound department, over a 2-year period (February 2017-January 2019).

Pregnancies were included if 1) both parents were aged over 18 years and unrelated (defined as first cousin or closer), 2) they were singleton, 3) the mother planned to have their baby at the SNM hospital and 4) the estimated due date could be calculated from last menstrual period or dating ultrasound. Those pregnancies where the fetus had a clinically obvious fetal structural or chromosomal abnormality were excluded. Women completed a questionnaire documenting their social (including nutritional), family, obstetric and medical histories

(including smoking, alcohol, chronic medical problems and medications). These data have already been reported by our group [21]. Only subjects who reported that their parents and grandparents (on both sides) were born in or close to Leh at altitudes>2,500m were considered as Ladakhi and thus recruited to the study.

Following birth and delivery of the placenta and umbilical cord, at least three 1ml aliquots of residual whole umbilical cord blood, with a separate serum sample, were frozen immediately at -80˚C. Information concerning the birth process (mode of delivery) and the neonatal characteristics routinely collected by the hospital (sex, weight, head circumference, crown-heel length and APGAR score) were recorded. The cord blood (and whole blood from their parents for future testing) was then shipped to Delhi and genomic DNA extracted at the Institute for Genomics and Integrative Biology, Delhi using commercial kits (Qiagen, Maryland, USA)and validated, standardized protocols. Common bi-allelic SNP genotyping was performed using Illumina Global Screening Array SNP-microarray technology (Illumina, San Diego, California, USA). Genotype assignment from the microarray fluorescence data was performed using Illumina's Genome Studio software.

We combined the genotypes of 316 Ladakhi individuals with reference individuals from surrounding populations living at high and low altitude. Human Genome Diversity project [22] datasets were included for the following different populations; Burusho, Han, Hazara, Indo Aryan, Japanese, Yakut Siberian. Similarly other population datasets included in this study were Munda [23], Sherpa [24], Tibetan [25] and Tibeto-Burman [26]. These individual cohorts were merged using autosomal SNPs found in common between all separate datasets. This merged dataset was then processed through a number of quality control steps using the software PLINK 1.9 [27, 28]. We included only individuals or SNPs that had < 5% missing genotypes, SNPs with a minor allele frequency (MAF) >1%, and SNPs with a HWE at significance of >1e$^{-6}$. Furthermore, we calculated closely related individuals using King [29], pruning one individual from a pair related by 2$^{nd}$ degree or closer. This left a final dataset of 1,413 individuals. Due to the variety of genotyping platforms between the different population references, the final set of common SNPs contained 60,280 markers.

We calculated principal components using PLINK [27], first pruning SNPs in linkage disequilibrium using PLINK's—indep-pairwise command using a window size of 1000, moving 50 SNPs, and using an r$^2$ threshold of 0.2 –leaving 41,718 SNPs in approximate linkage.

We conducted a genetic clustering analysis of Ladakhi populations with its neighboring populations using ADMIXTURE v1.2 [30], estimating individual ancestry proportions in an unsupervised analysis. The same cohort assembled in PCA was used for ADMIXTURE analysis, i.e., non-missing autosomal SNPs, individuals filtered for relatedness, and SNPs pruned for linkage disequilibrium. Unsupervised ADMIXTURE analysis was carried out over k = 2–4 populations.

We further investigated evidence of admixture within the Ladakh population's history by applying *f*-statistics [31] testing the strength of evidence that the Ladakh population is admixed between pairs of neighboring populations using the *f₃* implementation within *ADMIXTOOLS 2* (https://uqrmaie1.github.io/admixtools/index.html - a manuscript describing *ADMIXTOOL 2* is currently under preparation by its authors). The same cohort of individuals and SNPs that were used for PCA and ADMIXTURE analysis was used for *f₃*-statistic work.

We detected Runs of Homozygosity (ROH) [32] in Ladakhi individuals and compared population-averages to individuals from neighboring lowland and highland populations using PLINK's—homozyg command, with the following specific parameters;—homozyg—homozyg-window-snp 50—homozyg-snp 50—homozyg-kb 1500—homozyg-gap 1000—homozyg-density 50—homozyg-window-missing 5—homozyg-window-het 1. This was carried out on a subset of the individuals described in principal component, ADMIXTURE, and *f₃* analyses,

selecting individuals who were genotyped on SNP-microarray chips with an overlap of common SNPs >100K in number to faithfully detect ROH. This subset included individuals from the Indian Indo-Aryan, Ladakhi, Sherpa, Tibeto-Burman Bhutanese, Tibeto-Burman Nepalese, or Tibetan population labels. Common SNPs between these individuals were filtered according to the same parameters as in the PCA, leaving 117,044 common SNPs.

Genome-wide association tests were conducted using 601,887 genotyped SNPs. Linear regression was applied, implemented in PLINK v1.9 with the—linear command. We applied an additive genetic model adjusting sex and the first four principal components obtained from genome-wide SNP data. Gene variants associated with birthweight in the Ladakhi were compared to variants in the Global Biobank Engine [33] to confirm their association with birthweight in other populations.

Sample size calculations using algorithms incorporated in GWA power [34] were based on mean birthweights obtained from audit data collected in Leh prior to the study commencing. Calculations identified the study had an 80% power to detect a variant that would explain 11.4% (or greater) of the variability observed in birthweight given a discovery sample size of 300 neonates and alpha level of $5 \times 10^{-8}$.

## Results

In total, 316 eligible women were recruited and studied to delivery. Maternal and fetal baseline characteristics are summarized in Table 1.

Characteristics are summarized as mean [min/max] with birthweight of infant (kg) divided into those born at term gestations of >37 weeks and those born preterm at <37 weeks, birth weight centile (%) derived from Intergrowth, and those born at <10th centile recorded (defined as small for gestational age) and ancestry by history (Ladakhi and Tibetan).

Mean birth weight was 3.18kg (mean birthweight centile 44.5th) in term individuals (>37 weeks' gestation). According to intergrowth charts(https://intergrowth21.tghn.org), 14% of infants were born at less than the 10th centile (which defines a small for gestational age [SGA] newborn) and only 7.8% of all infants were born with a low birthweight (LBW, defined as less than 2.5kg). The average age for pregnant women was 28.9 years and 37.9% (120/316) were primigravida. All women reported 2, and most (96.8%) >3, generations of Ladakhi ancestry (as defined above). Ten women reported Tibetan ancestry: all gave birth at term, with a mean birth weight of 3.6 kg. Preterm birth was infrequent: 15/316 [4.7%] infants were born before 37 weeks, whose birth weight was, as expected, lower than those born at term (mean 2.4 vs 3.2kg, P = 0.0001] although birth weight centile was similar (52nd vs 45th respectively,

**Table 1. Maternal and fetal characteristics.**

| Characteristic | Term>37 <42+1 weeks n = 301 | Preterm<37 weeks n = 15 | Mean n = 316 |
|---|---|---|---|
| Maternal age (years) | 29.1 [18/40] | 26 [20/35] | 29.0[18/40] |
| Weight (kg) | 57.0 [37/92] | 56.9 [40/82] | 57.0[37/92] |
| Hb (g/dl) [min/max] | 12.7 [10.4/14.8] | 12.3 [7.5/17] | 12.3 (7.5/17) |
| Past Medical History (n) | Hypothyroid (6) | | |
| Dietary n (%) includes meat | 289 (96.3) | 14 (96) | 303 (96) |
| Primiparous n (%) | 111 (36.9) | 6 (40) | 117 (37) |
| Birth weight (kg) | 3.18[1.98–4.40] | 2.45[1.5–3.2] | 3.15[1.5–4.4] |
| Birth weight centile (%) | 44.5 [0.13/98.7] | 49 [1.5/99] | 44.9 [0.13/99.9] |
| <10th centile (%) | 42 (14) | 2 (13.3) | 44 (13.9) |
| Tibetan by history kg (centile) [n = 10] | | | 3.6 (70) |
| Ladakhi by history kg (centile) [n = 306] | | | 3.17 (43.9) |

p = 0.324) suggesting appropriate growth for gestational age. Recognizing that being preterm would confound birth weight if used in isolation, further analysis instead used birth weight centile as a more useful measure of growth potential, given that it adjusts for gestational age.

### Reconstruction of Ladakhi population history

We first sought the history and structure of the Ladakhi population from the perspective of genetics. We utilized our dataset of 316 Ladakhi neonates and compared it with other surrounding populations based on major language groups including Tibeto-Burman (Tibetans, Sherpa, Ladakhi), Indo-European (Indo-Aryan, Hazara), Austro-Asiatic (Munda) and isolate languages (Bursushaski) (Fig 1) using Principal Component Analysis (PCA). To understand the nature and extent of Himalayan ancestry in the Ladakhi population, we compared them with high altitude populations (Sherpa, Tibetans) and other Tibeto-Burman populations adjacent to the foothills of the Himalayas in Nepal and Bhutan.

In the PCA, the Ladakhi individuals project together, forming a genetic cluster that is distinct from the neighboring Indo-Aryan Indian population. This Ladakhi cluster forms a genetic cline between the two termini of principal component 1, which separates Indo-Aryan Indian individuals from individuals with Himalayan or East Asian ancestry. The second

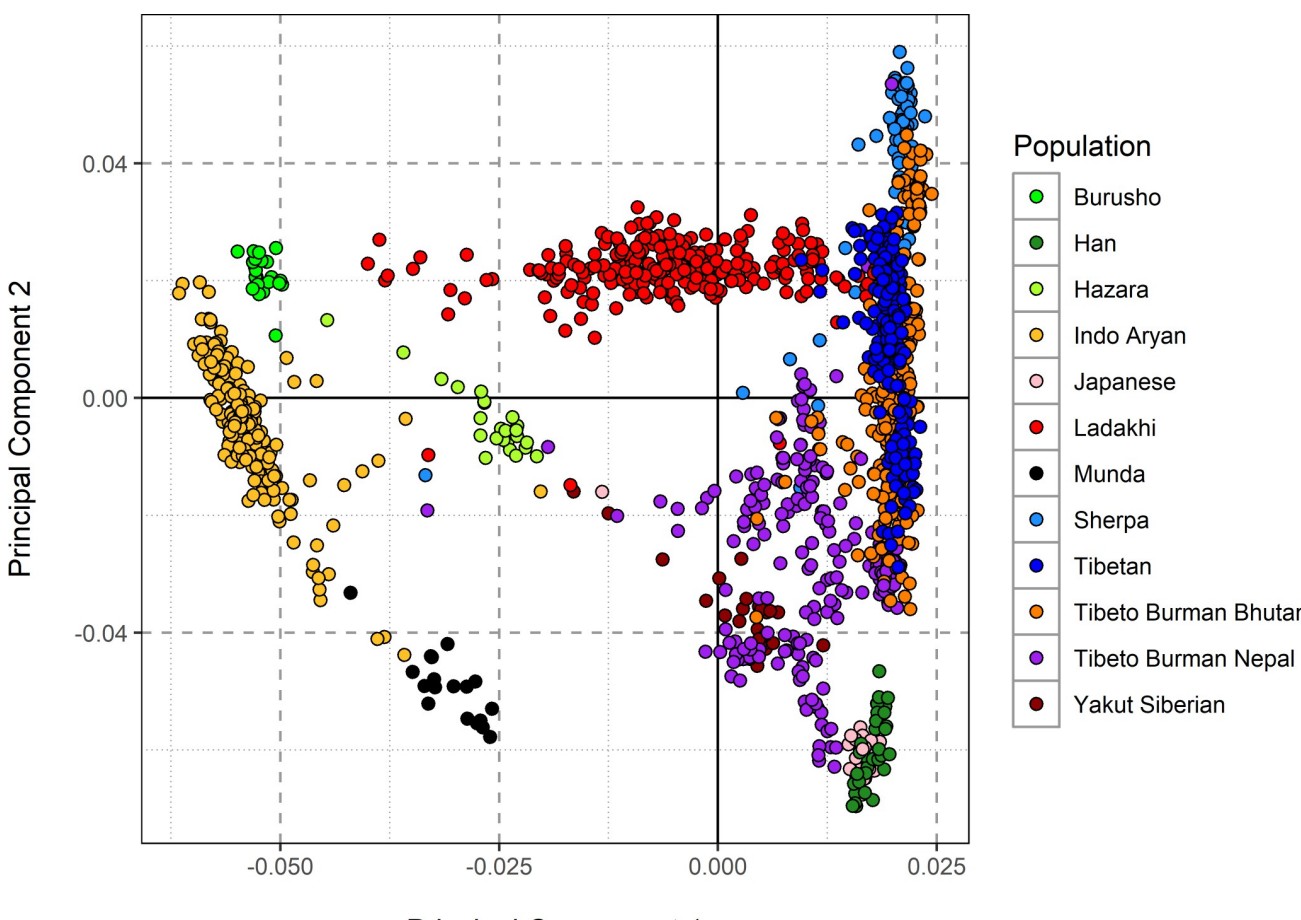

**Fig 1. Principal Component Analysis (PCA) of Ladakhi population.** PCA of Ladakhi and its neighboring populations are indicated in the figure legend. First and second components of PCA analysis with individual clusters labelled by different colours.

principal component appears to separate out Han (East Asian) individuals from Sherpa (high altitude), where Ladakhi individuals are placed towards the high-altitude-associated Sherpa population alongside individuals from Himalayan regions such as Tibet, Nepal, and Bhutan. The intermediate position of Ladakhi individuals on principal component 1 is suggestive of admixture between Indo-Aryan and Himalayan populations. The position of Ladakhi individuals on principal component 2 suggests that the Himalayan genetic group with which the Ladakhi share affinity is Tibetan or Sherpa, with no evidence in PCA of admixture with Han-related East Asian ancestries.

We further contextualized our population structure analysis using the maximum-likelihood estimation of individual ancestries using ADMIXTURE (Fig 2). At $k = 2$ the two ancestral components are maximized in either Himalayan or the East Asian Han populations (in red), and Indo-Aryan Indians (green). We observed that Ladakhi individuals are modelled with a slightly higher proportion of the East Asian ancestry than South Asian ancestral component, in agreement with their average location on principal component 1. At $k = 3$, one ancestry component, represented by blue, is maximized in the East Asian Han, the second (green) maximized in Sherpa, and the third (red) in Indo-Aryan Indians. The majority average component in Ladakhi individuals is the Sherpa-maximized ancestry (52.4%), then the Indo-Aryan-maximized component (32.3%), suggesting the majority of Ladakhi are closer to Tibeto-Burmans than Indo-Aryans. At $k = 4$ an ancestry component separates ancestry which is maximized in Tibetan-Bhutanese individuals from other East Asian sources. This component (in purple) is also found in appreciable proportions in other high-altitude populations. Ladakhi individuals also show appreciable proportions of this ancestry component common in both Nepalese Tibeto-Burman and Tibetans. However, the blue ancestry component at $k = 4$ in Ladakhi individuals continues to support admixture with other South Asian population such as Hazara, Burusho or Indo-Aryan Indians. Thus, Ladakhi appear genetically closer to Tibeto-Burman

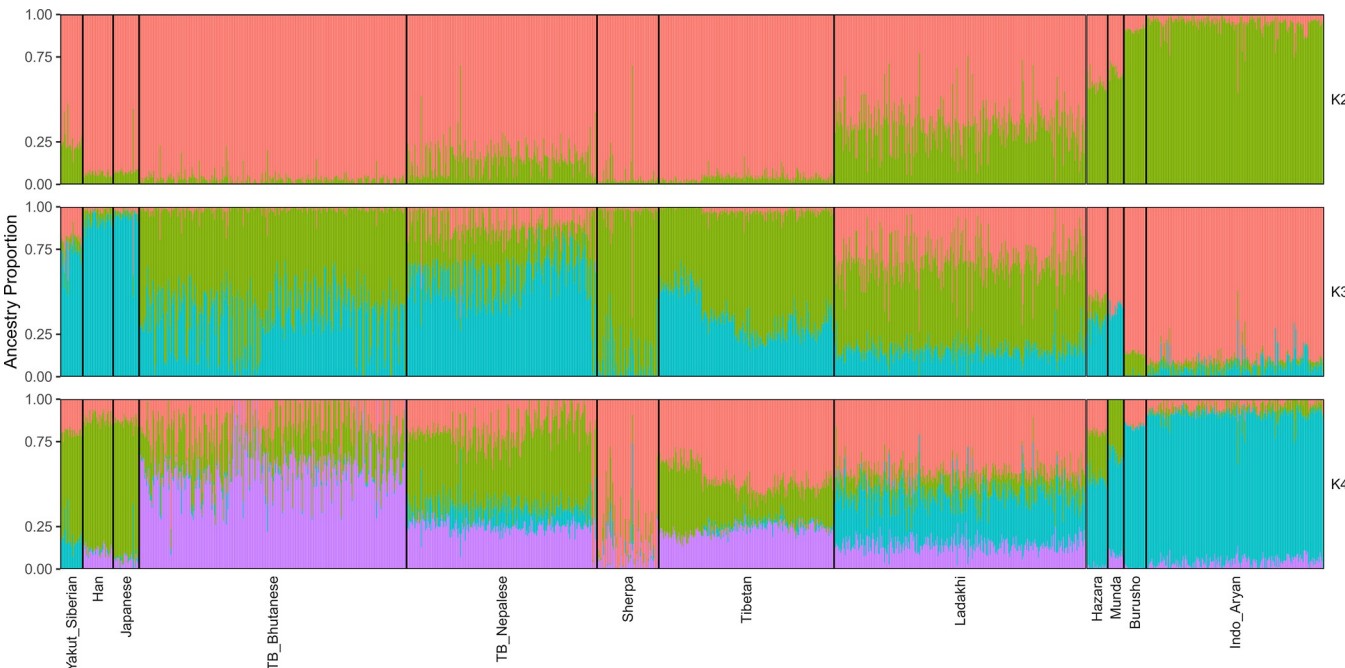

**Fig 2. ADMIXTURE analysis of the Ladakhi population.** ADMIXTURE analysis for K = 2 (top), K = 3 (middle) and K = 4 (bottom). ADMIXTURE analysis shows Ladakhi contain higher number of Tibeto-Burman ancestries and are relatively closer to Tibetans.

speakers of Himalayan region than Indo-Aryan populations of South Asia. Long-term isolation in remote highland, practices of endogamy culture and a founder effect might have added greatly to the formation of distinct genetic cluster in the Ladakhi agreeing with PCA result (Fig 1). In summary, the data supported Ladakhi as a genetically distinct Himalayan population closer to Tibeto-Burman population and Tibetan populations than to Indo-Aryan.

We next performed two additional analyses to further confirm that the Ladakhi are admixed between lowland Indo-Aryan and highland Tibeto-Burman sources. Firstly, we leveraged $f$-statistics [30] in the form of $f_3$(X; A, B) where X is tested for evidence of admixture between sources A and B, where a negative $f_3$-statistic is indicative of admixture. We placed Ladakhi as X, and tested combinations of lowland/Indo-Aryan populations (Burusho, Indo_-Aryan, Hazara, Munda) as X and highland/East or North Asian populations (Han, Japanese, Sherpa, Tibetan, TB_Bhutanese, TB_Nepalese, Yakut_Siberian) as Y, reporting those $f_3$ results with an absolute Z score >3 (Fig 3).

The strongest evidence of admixture (most negative $f_3$-statistic) is between Tibetan or Sherpa sources and Indo-Aryan or Burusho sources–supportive of results from PCA.

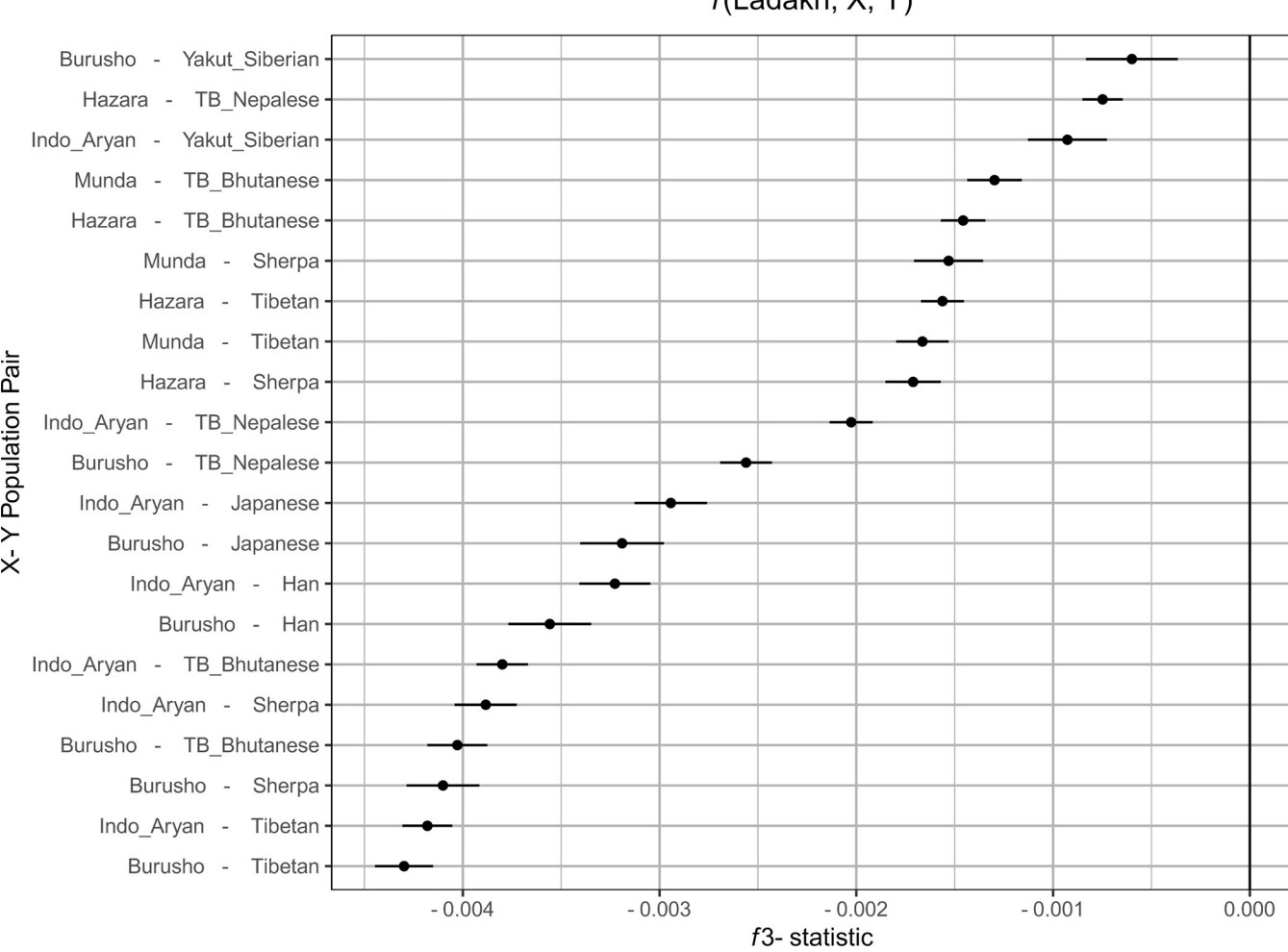

**Fig 3. Evidence of admixture in Ladakhi between lowland or Indo-Aryan populations and highland or north/east Asian populations measured using f3 outgroup statistics.** Along the x axis is the f3 statistic score. The more negative that statistic is, the greater is evidence of admixture. The error bars correspond to three standard errors of the estimated f3 statistic. Along the y axis are different pairs of putative source populations for an admixture event which creates the modern Ladakhi population.

Additionally, we performed Runs of Homozygosity detection using PLINK, comparing Ladakhi to a subset of neighbouring lowland and highland references. We detect elevated ROH (S1 Fig) in the high altitude Tibeto-Burman and Sherpa populations, agreeing with previous estimates [24], but only modest levels of ROH in Ladakhi more comparable to the general Tibetan or Indo-Aryan labelled individuals. These modest levels of ROH would be consistent with admixture between different ancestries.

### Birthweight analysis on Ladakhi subjects

In our analysis in reconstructing the genetic history of the Ladakhi population, we determined that our dataset represents a homogenous sample of genetic variation within the Ladakhi population. We therefore leveraged this homogenous sample of 316 Ladakhi neonates to determine if there were any genome-wide significant predictors of birthweight in the Ladakhi population, performing a GWAS of birth weight. We did not find any signals that were genome-wide significant after correction for multiple testing (i.e. $p < 5 \times 10^{-8}$) (S2 Fig). However, when we looked in the tail of the association statistics from our birthweight GWAS (i.e. $p = 1 \times 10^{-4}$ to $1 \times 10^{7}$ uncorrected), we noted the presence of multiple variants previously associated with haematological/cardiovascular traits of potential relevance to high altitude adaptation: rs16893892 in *RP5-874C20.3* associated with mean corpuscular hemoglobin, rs2298839 in *AFP* associated with mean platelet volume, rs9261425 in *TRIM31* associated with WBC count and rs362043 in *TUBA8* associated with atrial fibrillation [33].

Next, in order to determine whether the genetic architecture of birthweight in the Ladakhi population overlaps with that observed in lowland populations, we sought the association with birthweight in Ladakhis of the 70 genetic signals previously associated with birthweight in lowland populations [9]. Overall, 32 of these 70 signals were either genotyped or captured through linkage disequilibrium ($r2 > 0.8$) in our Ladakhi dataset. Seven of these 32 signals were significantly associated ($P < 0.05$, uncorrected) with birthweight in the Ladakhi GWAS study (Table 2). These variants mapped to five different birthweight associated genes (*ZBTB38*, *ZFP36L2*, *HMGA2*, *CDKAL1* and *PLCG1*) with the direction of effect being consistent with the original report [9] in all seven cases.

### Discussion

This study is the first to undertake a comprehensive fetal genotypic exploration of a high altitude population in relation to a birthweight phenotype. We showed that the Ladakhi are more closely related to Tibeto-Burman speaking populations than to the Indo-Aryan groups of

**Table 2. Single nucleotide polymorphism (SNP) of Ladakhi neonates compared with UK biobank.**

| SNP | Mapped Gene | SNP_ Annotation | P-values | | | Beta values | |
|---|---|---|---|---|---|---|---|
| | | | Ladakhi | UK biobank | GWAS_ Catalog | GWAS_ Catalog | Ladakhi |
| rs6440006 | ZBTB38 | Upstream | 0.03515 | $1.099e^{-5}$ | $4 \times 10^{-12}$ | -0.021224 | -0.0818 |
| rs8756 | HMGA2 | 3_prime_UTR | 0.000078 | $1.97e^{-138}$ | $1 \times 10^{-19}$ | -0.027582 | -0.176 |
| rs1351394 | HMGA2 | intronic | 0.000078 | $3.28e^{-137}$ | $2 \times 10^{-33}$ | -0.043 | -0.176 |
| rs7968682 | HMGA2 | intergenic | 0.000078 | $4.97e^{-130}$ | $4 \times 10^{-60}$ | -0.041831 | -0.176 |
| rs4952673 | ZFP36L2 | intergenic | 0.04 | NA-not available | $2 \times 10^{-11}$ | -0.020476 | -0.09315 |
| rs35261542 | CDKAL1 | intronic | 0.01864 | NA | $3 \times 10^{-45}$ | -0.040621 | -0.08755 |
| rs753381 | PLCG1 | missense variant | 0.02988 | NA | $3 \times 10^{-9}$ | -0.01512 | -0.0792 |

Single Nucleotide Polymorphism rs (rs) number and gene location along with P and Beta values

South Asia, providing clear evidence that Tibeto-Burman expansion occurred in North East India crossing the Himalayan range [35]. The majority of Ladakhi form a common clustering PCA, indicating the presence of a Ladakhi specific genetic ancestry component that is intermediate between Tibeto-Burman and Indo-Aryan ancestries. Analysis utilising *f*-statistics and ROH data further supports a demographic history of admixture between these sources and suggests a distinct demographic history in the Ladakh region, plausible due to its close proximity with the Tibetan plateau. Lack of large overlap between Ladakhi genotype data and existing references limited haplotype analyses of Identity-by-Descent segments or "ChromoPainter" based methods [36], which limited the power to analyse the demographic history of the population in more detail. Future work overcoming these technical limitations may provide further insights.

Individuals in Leh are born at a significantly higher birthweight than expected, based on existing literature for average birthweights in India at 2.8-3kg [37, 38] and seen from previous [21] studies. One such mechanism of protection of birthweight is through genetic selection as evidenced in other phenotypic traits of high altitude populations [18].

Birthweight is a complex trait with potentially conflicting parental inheritance patterns. As such, it is perhaps unsurprising that no one clear signal was identified at GWAS level, especially given the low numbers in the final study. Study of parental DNA and genetic signals in genes responsive to the hypoxia-responsive transcription factors *hypoxia inducible factors (HIFs)* in relation to offspring birth weight is an obvious follow up study to further explore HIF function and high altitude adaptation.

In relation to SNPs previously associated with birth weight by GWAS, that of rs1351394 of the *HMGA2* gene (encoding the high mobility group-A2 protein) was replicated in Ladakhi offspring. High-mobility group (HMG) proteins are ubiquitous nuclear proteins that bind to DNA and induce structural change in chromatin, thereby regulating gene expression [16]. *HMGA2* (the gene encoding the high mobility group-A2 protein) has been associated with human height and birthweight in lowland populations [39, 40] and with adipose mass variation in pigs [41], making it a biological plausible functional candidate.

*ZBTB38* encodes a zinc finger transcription factor that binds methylated DNA to enhance or repressing transcription in a way which is complex and likely cell-dependent [42]. Its expression appears to play a role in skeletal development [40]. Recent GWAS have reported SNPs in this gene to be associated with human height, perhaps through upregulation of insulin growth factor 2, a potent fetal growth factor [43].

A limitation of the study was the absence of genome wide significance in results presented. However, as for all adjustments for multiple comparison, many 'real' associations will fail to reach significance and thus we feel it reasonable to suggest that the higher-ranking alleles represent the best candidates for future study.

Our analysis was powered on a linear regression using quantitative traits identified as appropriate for the sample size selected, but we were unable to perform further sub-analysis of groups. We would have liked to have taken a case/control approach to the assessment of fetal growth but were underpowered to perform such analysis as $<3^{rd}$ centile at birth was rare. Only 14 infants were born <3rd centile. This would be a very interesting approach to apply to a larger sample size.

One of the strengths of this study relates to accurate dating of gestation from last menstrual period and early dating ultrasounds. This allowed for calculation of birth weight centile which adjusts neonatal weight for gestational age and sex at birth, allowing for better correlation with pathological poor growth. This study focused on fetal genotype, but it is recognized that maternal [44] and paternal genotype also has an effect. We have parental DNA and it would be beneficial in the future to analyze inheritance patterns of these fetal gene variants of interest.

## Conclusion

We found Tibetan ancestry in the Ladakhi population, and evidence of recent limited admixture with neighboring Indo-Aryan populations. The effects of a subset of variants known to predict birthweight in European-descent populations was replicated in the Ladakhi population. Replicated variants were focused on body anthropomorphic traits and hematological parameters. Further genomic study of high-altitude infants is indicated to validate these findings in conjunction with functional study to gain insight in potential mechanisms of action that may be amenable to therapeutic intervention.

## Supporting information

**S1 Fig. Runs of homozygosity(ROH) in different populations.** The plot shows the average total length of ROH in each of the populations. The coloured bar shows the arithmetic mean, and the box plots are normal box plots.
(TIF)

**S2 Fig. Manhattan plot showing GWAS results of birth weight genes across chromosomes.**
(TIF)

## Acknowledgments

This work would not have been possible without the tireless commitment of the staff and patients at Sonam Norboo Memorial Hospital, Leh Ladakh. The work has been supported by several hospital Medical Superintendents, in particular Dr Phunchok Wangchuk Koklukand by the Chief Medical Officer, Dr Motup Dorje.

## Author Contributions

**Conceptualization:** Padma Dolma, Hugh Montgomery, Dalvir Kular, David J. Williams, Sara L. Hillman.

**Data curation:** Sushil Bhandari, Mitali Mukerji, Bhavana Prasher, Dalvir Kular, Aniket Bhattacharyaa, Gianpiero L. Cavalleri, Sara L. Hillman.

**Formal analysis:** Sushil Bhandari, Bhavana Prasher, Edmund Gilbert, Gianpiero L. Cavalleri, Sara L. Hillman.

**Funding acquisition:** Hugh Montgomery, Vandana Jain, Vatsla Dadhwal, Sara L. Hillman.

**Investigation:** Padma Dolma, Bhavana Prasher, Hugh Montgomery, Dalvir Kular, Vandana Jain, Vatsla Dadhwal, Aniket Bhattacharyaa, Sara L. Hillman.

**Methodology:** Padma Dolma, Mitali Mukerji, Bhavana Prasher, Vandana Jain, Vatsla Dadhwal, Aniket Bhattacharyaa, Gianpiero L. Cavalleri, Sara L. Hillman.

**Project administration:** Padma Dolma, Vandana Jain, Sara L. Hillman.

**Resources:** Padma Dolma, Sara L. Hillman.

**Software:** Sushil Bhandari, Gianpiero L. Cavalleri.

**Supervision:** Mitali Mukerji, Bhavana Prasher, Hugh Montgomery, David J. Williams, Gianpiero L. Cavalleri, Sara L. Hillman.

**Validation:** Mitali Mukerji, Edmund Gilbert.

**Visualization:** Sushil Bhandari, Edmund Gilbert.

**Writing – original draft:** Sushil Bhandari, Sara L. Hillman.

**Writing – review & editing:** Sushil Bhandari, Mitali Mukerji, Bhavana Prasher, Hugh Montgomery, Dalvir Kular, Vandana Jain, Vatsla Dadhwal, David J. Williams, Aniket Bhattacharyaa, Edmund Gilbert, Gianpiero L. Cavalleri, Sara L. Hillman.

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
