## [Decision Letter · Decision Letter 0]

4 Jul 2022

PONE-D-22-15006Population History and Genome Wide Association Studies of Birth Weight in a Native High Altitude Ladakhi PopulationPLOS ONE

Dear Dr. Hilman,

Thank you for submitting your manuscript to PLOS ONE. After careful consideration, we feel that it has merit but does not fully meet PLOS ONE’s publication criteria as it currently stands. Therefore, we invite you to submit a revised version of the manuscript that addresses the points raised during the review process. Please ensure that your decision is justified on PLOS ONE’s publication criteria and not, for example, on novelty or perceived impact.

We look forward to receiving your revised manuscript.

Kind regards,

Giuseppe Novelli

Academic Editor

PLOS ONE

Journal Requirements:

Reviewers' comments:

Reviewer's Responses to Questions

**Comments to the Author**

1. Is the manuscript technically sound, and do the data support the conclusions?

Reviewer #1: Yes

Reviewer #2: Yes

Reviewer #3: Yes

2. Has the statistical analysis been performed appropriately and rigorously? 

Reviewer #1: Yes

Reviewer #2: Yes

Reviewer #3: I Don't Know

3. Have the authors made all data underlying the findings in their manuscript fully available?

Reviewer #1: Yes

Reviewer #2: Yes

Reviewer #3: Yes

4. Is the manuscript presented in an intelligible fashion and written in standard English?

Reviewer #1: Yes

Reviewer #2: Yes

Reviewer #3: Yes

5. Review Comments to the Author

Reviewer #1: There are a few misprints in the text (line 78, 84).

The expression 'mechanistic study' is philosophically charged. It would be necessary to explain why you consider genomic studies as 'mechanistic',

The absence of genome wide significance for the variants should deserve a deeper discussion, especially in regard to future studies.

Reviewer #2: In this study Authors evaluated the genetic protective role in maintain an adequate birthweight at high altitude. The subject is of interest the study well designed so I would like to congratulate with Authors for their effort

My comments are

1) according to recent obstetrical ultrasonography FGR is defined by a Delphi consensus (Gordjin UOG 2016) on the basis of severity of growth restriction (i.e. < 3 centile) or association of a birthweight < 10 and concomitant Doppler changes in fetal hemodynamics. Since I guess that Doppler data analysis are not available in the study population Authors should include a sub analysis of newborn with a birthweight < 3 to show whether their hypothesis fits also in such subjects

2)it is not clear how Authors controlled for other potential confounding variables that influence fetal growth such as alimentation, associated maternal diseases occurring in pregnancy, anemia, …

3)I suggest adding a table sub grouping women according to the ethnicity (Tibetan and Ladakhi) and performing an univariate analysis of maternal and obstetrical characteristics

4)the limitations of the study should be reported

Reviewer #3: The study is the first to analyze the genetic fetal growth protection at a high altitude.

The study is of great interest and is well designed. The Authors should be congratulated for this complex research.

Comments:

1. Since birth weight is an important parameter of the study, a better specification of Fetal Growth Restriction with ultrasound during pregnancy according to the Delphi consensus and its association with the birthweight could be included, if available.

2. Since fetal development is strictly linked to the maternal environment (nutrition, physical activity, individual and community stress) and not only to the altitude considered in the study and since mothers completed a comprehensive questionnaire of their pregnancy history, it would be interesting to make one or more tables showing the data.

It should include also the relevant medical history, when present.

3. It would be interesting to compare different ethnicities and maternal medical and obstetrical histories.

4. The limitations of the study should be better presented.

6. PLOS authors have the option to publish the peer review history of their article (what does this mean?). If published, this will include your full peer review and any attached files.

Reviewer #1: **Yes: **Bernardino Fantini

Reviewer #2: **Yes: **Giuseppe Rizzo

Reviewer #3: No

---

## [Author Response · Author response to Decision Letter 0]

3 Aug 2022

Our thanks to the reviewers for their insightful comments. Please find our responses below and reflected in the amended manuscript

Reviewer #1

Apologies. Misprints corrected 

The expression 'mechanistic study' is philosophically charged. It would be necessary to explain why you consider genomic studies as 'mechanistic'. 

Wording adjusted to better explain meaning 

The absence of genome wide significance for the variants should deserve a deeper discussion, especially in regard to future studies. 

Agreed and language amended including reference to future studies and more detail in discussion 

Reviewer #2

1) according to recent obstetrical ultrasonography FGR is defined by a Delphi consensus (Gordjin UOG 2016) on the basis of severity of growth restriction (i.e. < 3 centile) or association of a birthweight < 10 and concomitant Doppler changes in fetal hemodynamics. Since I guess that Doppler data analysis are not available in the study population Authors should include a sub analysis of newborn with a birthweight < 3 to show whether their hypothesis fits also in such subjects 

We now reference this definition (line 52) and the number of <3rd centile babies 

2)it is not clear how Authors controlled for other potential confounding variables that influence fetal growth such as alimentation, associated maternal diseases occurring in pregnancy, anemia, 

As part of the phenotype study we previously published associated features are reported (20). The majority of women studied were meat eaters, rather than vegetarian. Anaemia was not common (<10% of entire cohort) and women had very few co-morbidites (this information has been added to Table 1). Further data from our previous paper is added to Table 1 (line 108). 

We could not reliably measure all in the field and some (eg [Hb]) varies across pregnancy. Such issues will not confound, even if directly and mechanistically related to the gene variants under study, they will contribute to the observed genetic association but we have included in our discussion a line stating that introducing some environmental variation, may weaken the power of the study to identify genetic markers.

3)I suggest adding a table sub grouping women according to the ethnicity (Tibetan and Ladakhi) and performing an univariate analysis of maternal and obstetrical characteristics  

We had very few Tibetan women recruited in the overall study (n=10) (reported in Table 1) and those included in the birth weight analysis were of Ladakhi genomic ancestry by GWAS. 

4)the limitations of the study should be reported  

Thank you. We have made this more apparent within the body of text.

Reviewer #3

1. Since birth weight is an important parameter of the study, a better specification of Fetal Growth Restriction with ultrasound during pregnancy according to the Delphi consensus and its association with the birthweight could be included, if available.  

Please see comments above for reviewer 2 and we agree and have included in discussion 

2. Since fetal development is strictly linked to the maternal environment (nutrition, physical activity, individual and community stress) and not only to the altitude considered in the study and since mothers completed a comprehensive questionnaire of their pregnancy history, it would be interesting to make one or more tables showing the data. It should include also the relevant medical history, when present. 

Please see comments above for reviewer 2 and we agree and have included in discussion 

3. It would be interesting to compare different ethnicities and maternal medical and obstetrical histories.  Our study concentrated on Ladakhi pregnant women to better ascertain in this population genetic variants. The Number of women of different ethnicities was small in this study but would be interesting to review. Equally, the number of pregnant women recruited with significant medical/ obstetric complications was limited but would beef interest in the future. 

4. The limitations of the study should be better presented. Thank you. Discussion adjusted.

---

## [Editor Report · Decision Letter 1]

5 Aug 2022

Population History and Genome Wide Association Studies of Birth Weight in a Native High Altitude Ladakhi Population

PONE-D-22-15006R1

Dear Dr. Hillmann,

We’re pleased to inform you that your manuscript has been judged scientifically suitable for publication and will be formally accepted for publication once it meets all outstanding technical requirements.

Kind regards,

Giuseppe Novelli

Academic Editor

PLOS ONE
---

## [Editor Report · Acceptance letter]

7 Sep 2022

PONE-D-22-15006R1 

Population History and Genome Wide Association Studies of Birth Weight in a Native High Altitude Ladakhi Population 

Dear Dr. Hillman:

I'm pleased to inform you that your manuscript has been deemed suitable for publication in PLOS ONE. Congratulations! Your manuscript is now with our production department. 

Kind regards, 

on behalf of

Prof. Giuseppe Novelli 

Academic Editor

PLOS ONE